# The Significance of Xylem Structure and Its Chemical Components in Certain Olive Tree Genotypes with Tolerance to *Xylella fastidiosa* Infection

**DOI:** 10.3390/plants13070930

**Published:** 2024-03-23

**Authors:** Erika Sabella, Ilaria Buja, Carmine Negro, Marzia Vergine, Paolo Cherubini, Stefano Pavan, Giuseppe Maruccio, Luigi De Bellis, Andrea Luvisi

**Affiliations:** 1Department of Biological and Environmental Sciences and Technologies, University of Salento, Via Prov.le Monteroni 165, 73100 Lecce, Italy; erika.sabella@unisalento.it (E.S.); ilaria.buja@unisalento.it (I.B.); carmine.negro@unisalento.it (C.N.); luigi.debellis@unisalento.it (L.D.B.); andrea.luvisi@unisalento.it (A.L.); 2National Biodiversity Future Center, 90133 Palermo, Italy; 3WSL Swiss Federal Institute for Forest, Snow and Landscape Research, Zürcherstrasse 111, 8903 Birmensdorf, Switzerland; 4Department of Forest and Conservation Sciences, University of British Columbia, 3041-2424 Main Mall, Vancouver, BC V6T 1Z4, Canada; 5Department of Soil, Plant and Food Science, University of Bari “Aldo Moro”, 70126 Bari, Italy; stefano.pavan@uniba.it; 6Omnics Research Group, Department of Mathematics and Physics, University of Salento, CNR-Institute of Nanotechnology, INFN Sezione di Lecce, Via per Monteroni, 73100 Lecce, Italy; giuseppe.maruccio@unisalento.it

**Keywords:** xylem vessel anatomy, lignin, FISH-LSM, luteolin, salicylic acid, olive quick decline syndrome

## Abstract

Olive quick decline syndrome (OQDS) is a devastating plant disease caused by the bacterium *Xylella fastidiosa* (*Xf*). Exploratory missions in the Salento area led to the identification of putatively *Xf*-resistant olive trees (putatively resistant plants, PRPs) which were pauci-symptomatic or asymptomatic infected plants belonging to different genetic clusters in orchards severely affected by OQDS. To investigate the defense strategies employed by these PRPs to contrast *Xf* infection, the PRPs were analyzed for the anatomy and histology of xylem vessels, patterns of *Xf* distribution in host tissues (by the fluorescent in situ hybridization technique—FISH) and the presence of secondary metabolites in stems. The xylem vessels of the PRPs have an average diameter significantly lower than that of susceptible plants for each annual tree ring studied. The histochemical staining of xylem vessels highlighted an increase in the lignin in the parenchyma cells of the medullary rays of the wood. The 3D images obtained from FISH-LSM (laser scanning microscope) revealed that, in the PRPs, *Xf* cells mostly appeared as individual cells or as small aggregates; in addition, these bacterial cells looked to be incorporated in the autofluorescence signal of gels and phenolic compounds regardless of hosts’ genotypes. In fact, the metabolomic data from asymptomatic PRP stems showed a significant increase in compounds like salicylic acid, known as a signal molecule which mediates host responses upon pathogen infection, and luteolin, a naturally derived flavonoid compound with antibacterial properties and with well-known anti-biofilm effects. Findings indicate that the xylem vessel geometry together with structural and chemical defenses are among the mechanisms operating to control *Xf* infection and may represent a common resistance trait among different olive genotypes.

## 1. Introduction

*Xylella fastidiosa* (Wells) Raju (*Xf*) is on the “list of priority pests” in Europe [1] since it causes some severe diseases, some of which are olive quick decline syndrome (OQDS), Pierce’s disease in grapevine and citrus variegated chlorosis. This Gram-negative bacterium is among the vascular wilt pathogens colonizing the xylem and causing major decline in several host plants worldwide. 

Bacterial vascular pathogens can enter host tissues passively, via wounds or natural openings or, as for *Xf*, when they are delivered into the xylem by insect vectors, such as sharpshooter leafhoppers and spittlebugs. After its entrance in the host plants, *Xf* proliferates (degrading xylem cell wall components, parenchyma cells and pit membranes) and produces exopolysaccharides, i.e., viscous macromolecules that contribute to biofilm formation; as a result, the bacteria physically block water movement through xylem tissues, causing wilting symptoms [2]. 

A host may respond to vascular infection using various strategies, some of them species-specific and different in the host species, but one of the most common defense strategies is the induction of structural barriers to restrict the spread of the pathogen throughout the whole xylem. These barriers could act to limit vascular colonization vertically or horizontally. Vertical restriction is achieved mainly by tyloses and gels secreted by xylem parenchyma cells to create occlusions in the vessel lumen. The horizontal spread of pathogens is somehow prevented by the reinforcement of the walls through vascular coatings, mainly with lignin and suberin [3]. The anatomy of xylem vessels, for example, a thick cell wall, also plays a role in enhancing pathogen compartmentalization by these physical barriers [3]. 

As a long-lived tree species, the olive developed the production of specialized metabolites as sophisticated defense mechanisms [4]. In fact, the resistance of plants against vascular pathogens also occurs at the biochemical level through the production of secondary metabolites with antimicrobial properties such as phenolic compounds including secoiridoids (oleuropein), phenolic alcohols (hydroxytyrosol and tyrosol), flavonoids (e.g., apigenin, catechin, epicatechin, kaempferol, luteolin, myricetin, naringin and quercetin), hydroxybenzoic acids (e.g., gallic acid, protocatechuic acid, salicylic acid, syringic acid and vanillic acid) and hydroxycinnamic acids (e.g., coumaric acid, caffeic acid, ferulic acid, p-coumaric acid and sinapic acid) [4,5,6]. 

To date, there are some indications of common anatomical traits between resistant olive cultivars [7,8], but this information regards only a few agronomically important cultivars, Leccino and FS17 (also known as Favolosa). Also, even for the metabolites, no clear metabolomic reference profile is associated with resistant cultivars [9,10]. In any case, considerable information has been gathered on the metabolic profile of leaf extract, and only some limited knowledge about twigs is available. In particular, mechanisms controlling structural and chemical resistance in various *Xf* hosts are complex and, in olives, still need to be fully understood. 

To address this issue, a study with a multi-methodological approach was carried out, which included dendrochronological analyses of the wood, histochemical investigations, fluorescent in situ hybridization coupled with confocal microscopical analyses, and HPLC-MS analysis for the presence of secondary metabolites. This study was conducted on a genetic diversity panel of putatively resistant olive genotypes, having similar symptomatic profiles and different *Xf* concentrations, with the aim of understanding the plant response to *Xf* and determining if there is a common colonization pattern between the resistant genotypes.

## 2. Results

### 2.1. Vessel Diameter and Distribution

For each plant, both for the PRPs and for the symptomatic controls (CTRLs), the panorama image reconstruction of the radial sections simplified the identification of the xylem annual rings to proceed with the analysis of the vessel anatomical parameters by using the software WinCell^TM^ 2022.

The xylem vessels of the PRPs had an average diameter significantly lower than that of CTRLs in all annual rings (from 2016 to 2020) of the wood (Figure 1). 

More specifically, by classifying xylem vessels on the basis of the average diameters into four classes (<15 µm; 16–30 µm; 31–45 µm; 46–60 µm), the symptomatic CTRLs in all the five annual rings analyzed showed vessels predominantly in the class of diameter ‘31–45 µm’ while the PRPs, with a more heterogeneous distribution, had vessels falling mainly in the small-diameter class ‘16–30 µm’ (Figure 2).

However, for the vessel density (average number of vessels per square millimeter), no statistically significant differences were found between the PRPs and the CTRLs (Figure 3). Since the PRPs had smaller vessels, such as the resistant cultivar Leccino [11], the index of vulnerability to cavitation turned out to be lower, about half, than that in symptomatic CTRLs, with the exception of 2020, in which the two values were similar (Figure 4). The diameter of the xylem vessels positively correlated with the symptoms observed on the canopy (measured in Pavan et al. [12] through the adoption of a pathometric scale, Appendix A) (r = 0.53; *p*-value = 0.03 according to the coefficient index r of Pearson); therefore, as the average diameter of the vessels increased, the severity of symptoms observed on the canopy also became more evident. 

Likewise, the index of susceptibility to cavitation, indirectly calculated from the average diameter, positively correlated with symptoms (r = 0.76; *p*-value = 0.000939); in fact, a greater susceptibility to the cavitation phenomenon, in conditions of water stress also induced by the *Xf*, can contribute to the aggravation of the symptomatologic picture.

### 2.2. Histological Characteristics of the Xylematic Vessels

By means of histochemical staining, it was possible to highlight some differences in the structural composition of the xylem vessels of the PRPs in comparison with the symptomatic CTRLs.

In relation to the cellulose distribution, no statistically significant difference was found, whereas the average intensity of the lignin-associated fluorescence signal was found to be significantly higher in PRPs compared to the CTRLs (Figure 5). In particular, in the PRPs, a greater quantity of lignin was observed in the parenchyma cells of the rays of the wood (Figure 5); these cells are arranged radially in the cross section of the wood and have the function to transversely connect the wood tissues. 

### 2.3. FISH-LSM Analysis of Branches

The 3D images obtained from the image modeling of the FISH-LSM analysis showed, as expected, a massive presence of bacteria aggregates in the xylem vessels of CTRLs; the aggregates were distributed according to a classic radial pattern (Figure 6). On the contrary, in xylem vessels of the PRPs, bacteria mostly appeared as individual cells or as small aggregates, meaning that the vessels were not completely occluded (Figure 6). This could explain how some plants such as SX_32 (cluster K2 with Tunisian cultivars) and SX_67 and SX_2 (both grouped in the sub-cluster K1/L including the *Xf*-resistant cultivar Leccino), despite having an *Xf* concentration comparable to those of the susceptible plants (10^5^ cfu/mL), did not show symptoms, with a disease severity score equal to 0.00 (no symptoms; Appendix A). 

An interesting aspect observed in the sections of the PRPs is that the bacterial cells were often included in the autofluorescence signal of gels and phenolic compounds associated with the xylem tissue (Figure 7) and everted into the lumen of the vessel (a mechanism that leads to tylosis formation; Appendix A).

### 2.4. Analyses of Secondary Metabolites in Olive Stems

Concerning secondary metabolites, the concentration of luteolin in the stem tissue of the PRPs was on average 7–14 times higher in comparison with the symptomatic CTRLs (Figure 8A). In particular, the genotypes with the higher luteolin concentration were SX_32 (clustered with Tunisian cultivars, cluster K2) and SX_67 of the sub-cluster K1/L (related to the cultivar Leccino) (Figure 8A). On the contrary, the other flavonoid with a statistically significant difference among CTRLs and PRPs was naringenin, which showed an opposite trend: in the PRPs, it was not detectable (Figure 8B), while in the symptomatic CTRL plants, it was present in the range of 0.72 and 2.58 µg/ mg fresh weight (Figure 8B). Stem tissues of the PRPs contained on average 8 times more salicylic acid than the CTRLs (Figure 8C). On the contrary, the tyrosol level was higher (from 8 to 72 times) in the CTRLs when compared with the PRPs (Figure 8D).

## 3. Discussion

Vascular pathogens cause among the most destructive plant diseases; they live deep inside their host plants and, for this reason, the measures of control for this group of pathogens are ineffective and the studies to understand the biology of these diseases are complicated. Now, the most effective control strategy is the use of genetic resistance and, so, the comprehension of the molecular mechanisms and the pathogen–host plant interactions is decisive. 

In this work, we studied the defense strategies in ten putatively resistant olive plants belonging to different genetic clusters, positive in the diagnostic test for *Xf* but with no symptoms on the canopy, despite being cultivated in orchards severely affected by the pathogen. To elucidate the defense strategies carried out, xylem vessels were examined from several points of view. The anatomical analyses confirmed that the PRPs were characterized by small-diameter vessels compared to the symptomatic plants (CTRLs), which displayed a higher number of wide-diameter vessels (Figure 1 and Figure 2). These data are in accordance with previously published works which reported the role of small vessel diameter as a factor increasing the plant’s resistance to *Xf*, both in resistant olive cultivars [7,8,13] and in other *Xf*-resistant host species [14]. In the analyzed plants, no statistically significant variation was found in the vessel frequency (Figure 3); in any case, this anatomical parameter shows high mean sensitivity to environmental conditions, and no correlation with *Xf* resistance has yet been demonstrated [15,16]. The main hypothesis on the role played by xylem geometry in *Xf* resistance is linked with the susceptibility to cavitation which, in infected plants, cooperates in worsening the symptomatology [7,17,18]. In fact, the indices of vulnerability to cavitation calculated in the PRPs and in the symptomatic plants (Figure 4) confirmed that the PRPs seemed to be less susceptible to cavitation, and the values of the indices of vulnerability to cavitation in the symptomatic plants significantly correlated with the severity of the symptoms (r = 0.76; *p*-value = 0.000939). 

Further emerging evidence is the defense role played by lignin not only against other vascular phytopathogens but also for *Xf* [9,11,19,20,21]. The histochemical staining achieved in this study revealed lignin accumulation in the PRPs with a specific distribution in the parenchyma rays (Figure 5); these cells are arranged radially in the cross section of the wood and have the function of transversal connection between the tissues that form the stem. The role of these cells in the defense against vascular pathogens is known. The main mechanism of defense concerns the CODIT model (Compartmentalization of Decay in Tree [22]), a model according to which the plant creates physical barriers to compartmentalize the pathogen and confine it, avoiding its systemic spread in the host. Based on the role that these barriers can play in blocking the movement of the bacterium, the PRPs were also subjected to FISH-LSM analyses with probes specific for *Xf* that confirmed that in the xylem vessels of the PRPs, *Xf* appeared mostly as individual cells with a random distribution and not the classic radial patterns among adjacent vessels [23]; *Xf* cells appeared to be incorporated in the autofluorescence signal of gels and phenolic compounds associated with the xylem tissue (Figure 7). This evidence indicates the involvement, in the *Xf* resistance mechanism, not only of physical barriers but also of chemical ones. Also, for Pierce’s disease in grapevine, individual *Xf* cells or small aggregates have been observed when *Xf* adheres to the tyloses; in that case, it has been suggested that tyloses, as well as acting as a physical barrier, may supply a desirable surface for *Xf* colonization, which is perhaps justified by the presence of by-products of cell wall turnover associated with tylose development [24]. As reported previously [25], the primary wall which expands to form the tyloses (extensions of the protective layers of the parenchyma) secretes substances such as gels and phenols to make these physical/chemical barriers more effective [3]. 

Phenolic compounds are important substances which, in the vascular system of plants, act not only as a physical barrier but also carry out direct antimicrobial activity [26]. The specific composition of vascular deposits varies based on the particular host–pathogen interaction [3]. Most of the works in the literature describe the phenolic composition of olive cultivars, also in relation to *Xf* infection, by analyzing leaves and petioles [6,9,10]. Specifically, in the work of Vergine et al. [10], HPLC ESI/MS-TOF analyses were performed on leaf petiole extracts of thirty putatively resistant genotypes. Through uni- and multi-variate statistical methods, the authors characterized the metabolic profiles of olive genotypes genetically related to the cultivars “Leccino” and “Ciciulara”, but no common metabolic pattern was found between the selected genotypes versus the susceptible one. In our work, we focused on the same tissues (stem) studied for anatomical/histochemical characteristics and for the bacterial pattern distribution. The phenolic compounds that showed statistically significant differences among the stems of asymptomatic PRPs and symptomatic plants were the flavonoids luteolin and naringenin and the hydroxybenzoic acids salicylic acid and tyrosol (Figure 8). These phenolic profiles, obtained by analyzing the stems, showed a different pattern of phenolic composition if compared with the same plants evaluated by processing the leaves in a previously published study [10]; this could suggest a tissue-specific phenolic pattern. Luteolin was significantly more abundant in the PRPs (Figure 8). Several reports showed that luteolin and its derivates have anti-bacterial [27,28] and anti-biofilm [29,30,31] properties. Also, the in vitro antimicrobial activity of the luteolin was tested, and inhibitory activity against the *X. fastidiosa* Salento-1 isolate has already been demonstrated [32]. Naringenin is the first intermediate molecule of flavonoid biosynthesis and has roles in many aspects of plant physiology with tissue-specific properties [33]. In some herbaceous plants, naringenin is involved in salt/osmotic and water stress [34,35,36]. In wood, the ability of naringenin to inhibit 4CL (4-Coumarate: CoA ligase) activity in lignifying xylem has been reported [37]. Therefore, its significantly low content in stem tissues of the PRPs (Figure 8) can be linked to the lignin increase found in the xylem vessels as a defense strategy. Among the hydroxybenzoic acids, salicylic acid is well known to play an important role in plant stress response; in the PRPs, a significant increase in salicylic acid was reported compared with the symptomatic plants (Figure 8). Salicylic acid is crucial to induce the expression of pathogenesis-related genes [38] and the synthesis of compounds which are involved both in local and systemic acquired resistance in plants [39]. Moreover, a wide range of studies reported salicylic acid as a pivotal signaling molecule that plays a crucial role in plant tolerance and resistance both to biotic and abiotic stresses by activating mechanisms that act at different levels [40,41], some of which are the induction of activities and the expression of antioxidant enzymes such as superoxide dismutase (SOD), catalase (CAT) and peroxidase (POD), thus enhancing a plant’s ability to eliminate reactive oxygen species (also ROS produced as by-products by plant-pathogenic microbes) and reducing cellular oxidative damage [42,43,44]. Salicylic acid acts as a fundamental signal for SAR (systemic acquired resistance) [45] and it induces several pathways (such as the MAPK, mitogen-activated protein kinase and the CDPK, calcium-dependent protein kinase), playing a central role in plant–pathogen interaction [46]. Salicylic acid interacts with other hormones and acts with other signaling molecules such as jasmonic acid (JA), auxin and ethylene (ETH) to regulate plant tolerance to stress [47]. Another mechanism is the induction of active anti-pathogenic substances biosynthesis [48,49]. In addition, salicylic acid can induce cell wall strengthening as a resistance response to plant disease [50]. Vañò et al. [51] supported the key role of the salicylic acid in plant defense, suggesting that it could play a significant role in the response to the presence of *X. fastidiosa* in the vessels. On the other hand, the unexpected significant increase in tyrosol in symptomatic plants (Figure 8) may be related to the water stress suffered by symptomatic plants, as indicated by Mechri et al. [52], who reported that the concentration of tyrosol was significantly increased in olive trees under water-stressed conditions.

## 4. Materials and Methods

### 4.1. Plant Materials

Putatively resistant plants (PRPs) previously identified during exploratory surveys in the autumn of 2019 and 2020 in olive orchards heavily affected by *Xf* were used as the starting material for this study. According to the genetic profile assessed by Pavan et al. [12] with a selection of SSR markers, the PRPs were included in two genetic clusters: K1 (including Italian cultivars among the *Xf*-resistant Leccino and FS17) and K2 (including Tunisian cultivars). Cluster K1 was composed of the two sub-clusters, K1/L (genotypes closely related to Leccino) and K1/C (genotypes closely related to the cultivar Ciciulara). Ten of these PRPs, i.e., SX_32 (belonging to the K2 genetic cluster) and other individuals grouped in cluster K1 (SX_31, SX_33, SX_5, SX_29, SX_27, SX_25, SX_26, SX_67 and SX_2) were analyzed in this work. As reported in Pavan et al. [12], the selected PRPs displayed a disease severity index < 1.5 (Appendix A), based on the 0–3 symptom scale proposed by Luvisi et al. [6], and different *Xf* concentrations after TaqMan real-time PCR protocol analysis [53,54]. In addition, olive trees (disease severity index ≥ 1.5) belonging to the *Xf*-susceptible cultivar “Cellina di Nardò” were also collected in every area in which each PRP was identified. Therefore, each PRP was compared to a CTRL sample chosen in the proximity of the selected plant, representing the *Xf* infection status of the explored area.

More detailed information about PRPs and CTRLs samples is reported in Appendix A.

### 4.2. Wood Anatomy

Five-year-old branches (*n* = 3 per plant) from PRPs and CTRLs were sampled and processed according to Sabella et al. [16]. A 4% aqueous formaldehyde solution was used as an antiseptic and fixative [55]. Radial sections were obtained from branches previously cut into pieces approximately 2 cm long. Thin perpendicular sections (13–15 μm thick) were cut from each piece with a sliding WSL Lab microtome (WSL Institute, Birmensdorf, Switzerland) (modified Reichert-type [56]). A safranin and astrablue mixture (50:50) was used to stain the sections; then, they were dehydrated with ethanol (75%, 96%, and 100 %), and after a wash with xylol, they were finally embedded in Canada balsam [57]. Images (5X magnification) were taken using a digital camera (EC3, Leica Microsystems, Wetzlar, Germany) connected to a transmitted light microscope (Orthoplan, Ernst Leitz, Wetzlar, Germany). For each thin section, several images were obtained and later used to compose a single image using the PTGui software 12.8 (New House Internet Services B.V., Rotterdam, NL, USA) [58]. Then, for each year (2016 to 2020), the diameter of xylem vessels (µm) and the vessel distribution (N/mm^2^) were determined using the image analysis software WinCell^TM^ 2022 (Regent Instruments Inc., Québec, QC, Canada). The vessel diameter and the vessel distribution were computed according to Scholz et al. [59]; the mean diameter of xylem vessels was obtained by calculating the mean diameter from 100 vessels for each individual and, according to the data, xylem vessels were divided into six classes of diameter (0–15 µm; 16–30 µm; 31–45 µm; 46–60 µm; 61–75 µm; 76–90 µm).

To evaluate the correlation between the mean diameter of the xylem vessels and the disease severity data of the canopy of the PRPs and of the CTRLs, the index r of Pearson was calculated by using the open-source software R ver.3.3.2 (R Foundation for Statistical Analysis).

Vulnerability to cavitation (vulnerability index, VI) was calculated using the equation proposed by Carlquist S. [60], as follows:VI = VD/VF
where VF is the vessel frequency (number of vessels per mm^2^, N/mm^2^), while VD is the mean vessel diameter (µm).

### 4.3. Histological Methods

Segments of 1.5 × 1.5 cm were excised from one-year-old branches with sterile razor blades and processed according to Cardinale et al. [23]; surface sterilization was obtained by using 70% ethanol for 1 min, followed by three rinses in sterile distilled water. The sample segments were fixed in 4% paraformaldehyde in phosphate-buffered saline (PBS 1×); after the incubation overnight at room temperature, they were washed with PBS buffer for 10 min at room temperature. The samples were then dehydrated (two 1 h incubations in each of 70%, 80%, 95% and 100% ethanol) and embedded in paraffin; 40 µm thick sections were obtained with a microtome Leica RM 2155 (Leica Microsystems, Mannheim, Germany). Sections were stored in a solution 1:1 (*v*/*v*) PBS:96% ethanol at −20 °C until staining. To dissolve the paraffin, sections were incubated in toluene for 3 min at 43 °C. Then, the sections were washed twice for 5 min each with PBS buffer and stained with Calcofluor white M2R and Safranin O, to detect, respectively, cellulose and lignin deposition. Stain solutions were prepared as follows: Calcofluor White M2R (Sigma-Aldrich, Milan, Italy) 0.02% (*w*/*v*) in distilled water and Safranin O (Sigma-Aldrich, Milan, Italy) 0.5% (*w*/*v*) dissolved in 50% EtOH. Calcofluor White M2R staining was achieved according to Mitra and Loqué [61]. Sections were excited at 405 nm and images were taken using a filter BP 420–480 nm. Safranin O staining was carried out following the protocol of Bond et al. [62]. Sections were then excited at 488 nm and images obtained by using a filter BP 500–700 nm. All images were captured with a confocal laser-scanning microscope (Carl Zeiss LSM 700 laser scanning microscope, Jena, Germany); the same microscope settings were maintained for all the images which were examined by using the Zeiss LSM image examiner software 4.2.0.121.

### 4.4. FISH-LSM Analysis

Sections were obtained as reported in the histological methods and processed according to Cardinale et al. [23]. After the toluene removal and two rinses for 5 min in PBS buffer, sections were permeabilized for 10 min at room temperature in 0.5 mg mL^−1^ lysozyme (Life Technologies Italia, Milan, Italy).

After dehydration (ethanol 96–70–50%, 3 min each), hybridization was carried out at 42 °C for 120 min with the Cy3-labeled *X. fastidiosa*-specific KO 210 probe, followed by 10 min of washing at 43 °C [23]. Ice-cold water was used for a final rinse, and the Citifluor AF1 antifade reagent (Linaris Biologische Produkte GmbH, Dossenheim, Germany) was used as an antifade reagent for the montage onto the glass slide. Finally, a coverslip was carefully placed on the sections. The slides could be stored for up to 4 days in the dark at 4 °C. A confocal laser-scanning microscope (Carl Zeiss LSM 700 laser scanning microscope, Jena, Germany) was used for the observations. Cy3 was excited with the laser line 561 nm; the laser lines 405 and 488 nm were used to induce the autofluorescence of tissues and of gel and phenolic compounds. Emissions were captured in the range of 570–613 nm for Cy3; 420–480 nm for the plant autofluorescence; and 500–700 for the gel/phenolic compounds. Then, 20–50 optical slices were acquired with the Z-stack tool, and the resulting “confocal stacks” were analyzed to obtain the three-dimensional reconstructions with the Zeiss LSM image examiner software v. 4.2.0.121.

### 4.5. Stem Tissue Processing for Secondary Metabolites Analysis

Segments of one-year-old branches were ground in pre-cooled mortar to a fine powder using liquid nitrogen and processed according to Nutricati et al. [38]. Approximately 0.5 g of pulverized tissue was homogenized in 3 mL of 100% methanol (1:6). The obtained extracts were sonicated for 60 min; then, they were centrifugated for 15 min at 10,000× *g* to remove cellular debris. The supernatant was divided into two equal portions which were dried at 40 °C with a flow of nitrogen. One half of the dried sample was resuspended in 900 µL of 50 mM sodium acetate (pH 4.5) and 100 µL of water containing 10 U of almond β-glycosidase (EC 3.2.1.21) (Sigma Aldrich, Milan, Italy). The other half was resuspended in 900 µL of 50 mM sodium acetate (pH 4.5) and 100 µL of water. The reactions were kept overnight at 37 °C and stopped with 75 µL of 70% perchloric acid (5% (*v*/*v*) final concentration); the reactions were stored at 4 °C for 1 h. Polymers were removed with centrifugation at 14,000× *g* for 15 min, and the obtained supernatants were extracted with 2.5 mL of cyclopentane/ethyl acetate (1:1, *v*/*v*). The organic upper phase was dried at 40 °C under a flow of nitrogen. Then, 200 µL of methanol was added to the residues and filtered through 13 mm nylon 0.45 µm syringe filters (Waters S.p.A, Milan, Italy) prior to HPLC analysis. Next, 20 µL (from the final 200 µL methanolic extract) was analyzed with an Agilent 1200 liquid chromatography system (Agilent Technologies, Palo Alto, CA, USA) equipped with a standard autosampler [11]. The high-performance liquid chromatography (HPLC) column used was an Agilent Extend-C18 (1.8 µm, 2.1 by 50 mm). Separation was performed at room temperature with a linear gradient of 1% (*v*/*v*) acetic acid to 100% methanol at a flow rate of 1 mL/min. Next, 100% methanol was used to wash the column for 10 min, which was then equilibrated again with 1% acetic acid in water for 10 min. Commercial standards (tyrosol, luteolin, naringenin, salicylic acid, caffeic acid, quercetin and gentisic acid) (Sigma-Aldrich, Milan, Italy) were used for the standard calibration graphs used for the quantification of the compounds from olive stems. The limit of quantification (LOQ) was assessed as the signal-to-noise ratio of 10:1 and the limit of detection (LOD) as the signal-to-noise ratio of 3:1. Intra-day and inter-day precision was used to evaluate the repeatability of the methods and was expressed by the relative standard deviations (RSDs). An olive stem extract was injected (*n* = 5) on one day (intra-day precision) for three consecutive days (inter-day precision, *n* = 15).

## 5. Conclusions

The *Xf* resistance response characterized in the identified asymptomatic olive genotypes has many points in common with those already found in the resistant cultivar Leccino, especially regarding the role of the xylem vessel geometry and structural composition as physical barriers. In addition to these strategies, our analysis identified other chemical barriers presumably operating with the physical ones to make the defense strategy more effective. Surely, additional experiments in controlled conditions and with artificial bacterial inoculation need to be conducted to evaluate the constitutive or induced responses to pathogen infection in the olive genotypes reported here. In the host plant, many mechanisms and molecules are involved in the *Xf* response, including the endophytic microbiota [63]. Our data are a contribution towards understanding the mechanisms of resistance to *X. fastidiosa*; among the highlighted features, salicylic acid’s role will need to be further explored. Understanding the mechanisms underlying plant disease resistance is of crucial importance, mainly because the research on plant resistance is still limited to a restricted number of pathosystems. In this context, our work appears to be a fundamental starting point and a compass to guide investigations at a genetic and molecular level; the next step should be the identification of the genetic basis underlying the mechanisms and molecules involved in the response to *Xf*. Furthermore, such resistance mechanisms could also be useful against other vascular pathogens for more efficient and effective utilization in crop improvement and protection.

## Figures and Tables

**Figure 1 plants-13-00930-f001:**
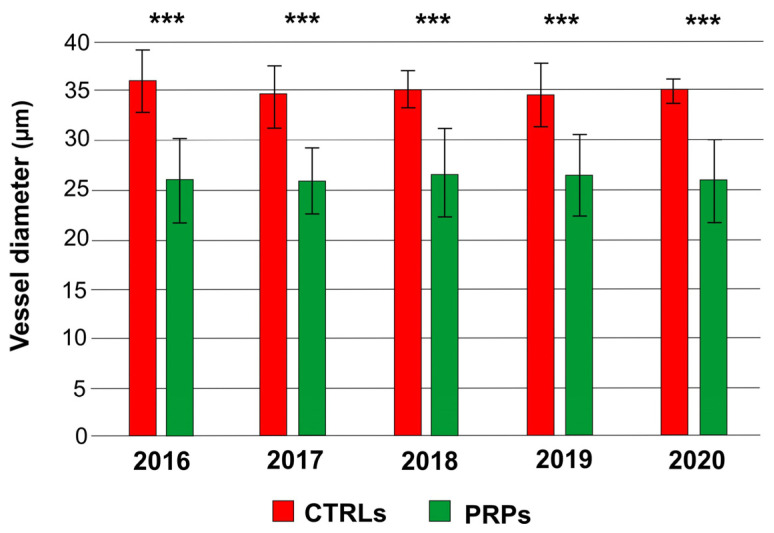
Mean diameter of xylem vessels in symptomatic control plants (CTRLs) and in the putatively resistant plants (PRPs) for each annual ring (2016, 2017, 2018, 2019, 2020). Statistically significant differences are highlighted according to Student’s *t*-test (*** *p*-value < 0.001).

**Figure 2 plants-13-00930-f002:**
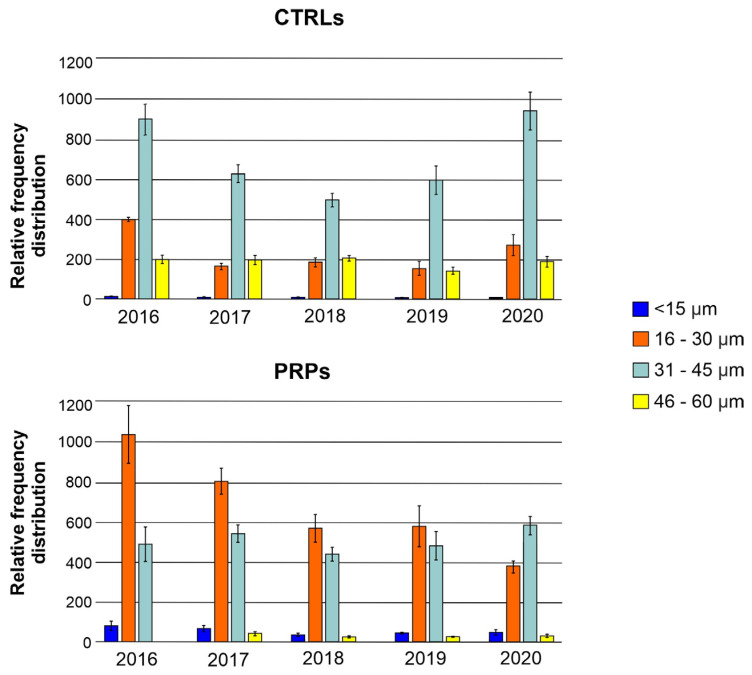
Relative frequency distribution of xylem vessels by diameter class (<15 µm; 16–30 µm; 31–45 µm; 46–60 µm) for each annual ring (2016, 2017, 2018, 2019, 2020) studied in wood in symptomatic control plants (CTRLs) and in the putatively resistant plants (PRPs).

**Figure 3 plants-13-00930-f003:**
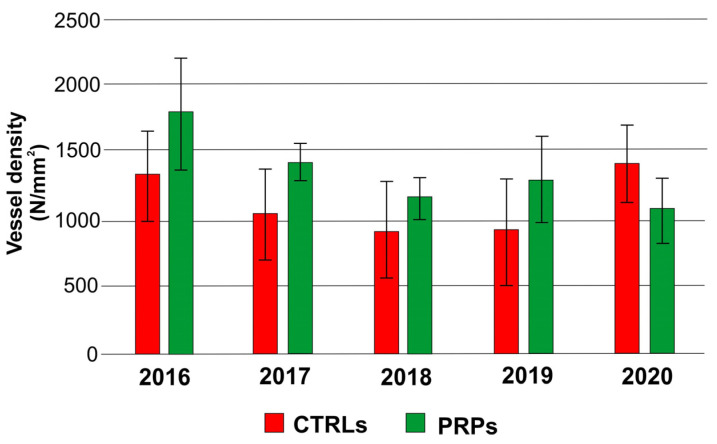
Vessel density (N, number of vessels per mm^2^) in the symptomatic control plants (CTRLs) and in the putatively resistant plants (PRPs) for each annual wood ring (2016, 2017, 2018, 2019, 2020).

**Figure 4 plants-13-00930-f004:**
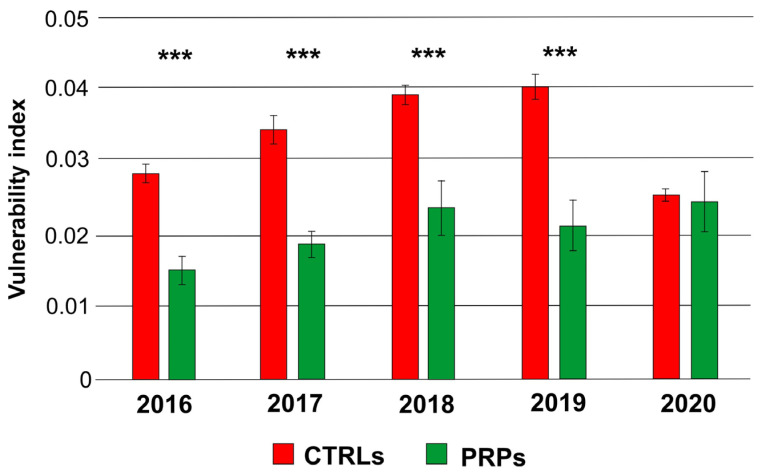
Index of vulnerability to cavitation in symptomatic control plants (CTRLs) and in putatively resistant plants (PRPs) for each analyzed annual wood ring (2016, 2017, 2018, 2019, 2020). Statistically significant differences are highlighted according to Student’s *t*-test (*** *p*-value < 0.001).

**Figure 5 plants-13-00930-f005:**
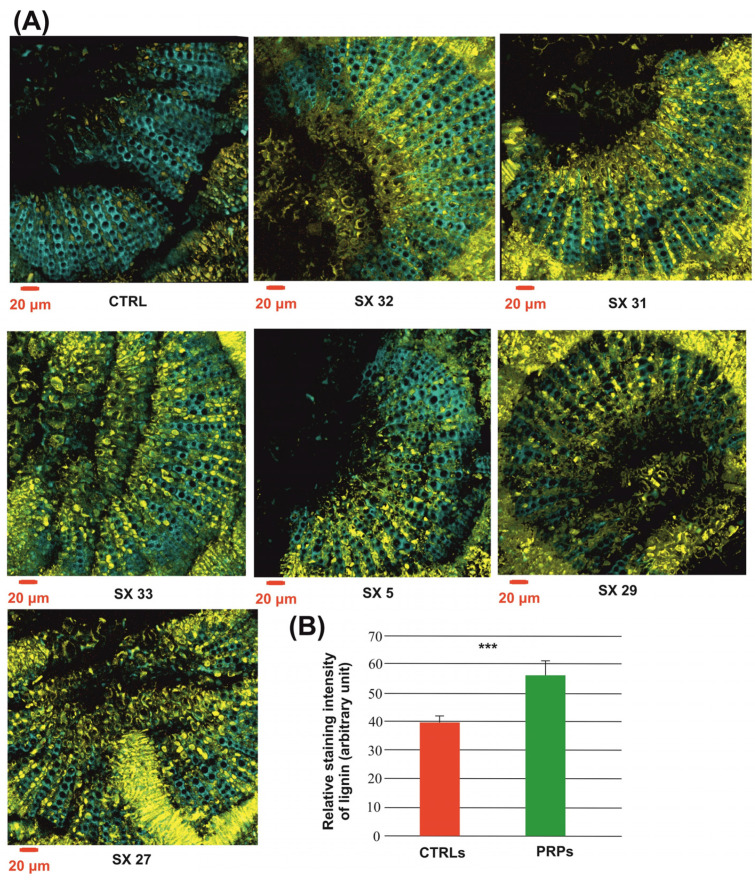
One-year branch sections stained with Calcofluor White M2R and Safranin O for setting in evidence cellulose (cyan) and lignin (yellow), respectively. (**A**) CTRL: representative image of symptomatic controls; SX_n_: representative images of the putatively resistant plants (PRPs). (**B**) Fluorescence index of lignin stained with Safranin O. *** *p*-value < 0.001.

**Figure 6 plants-13-00930-f006:**
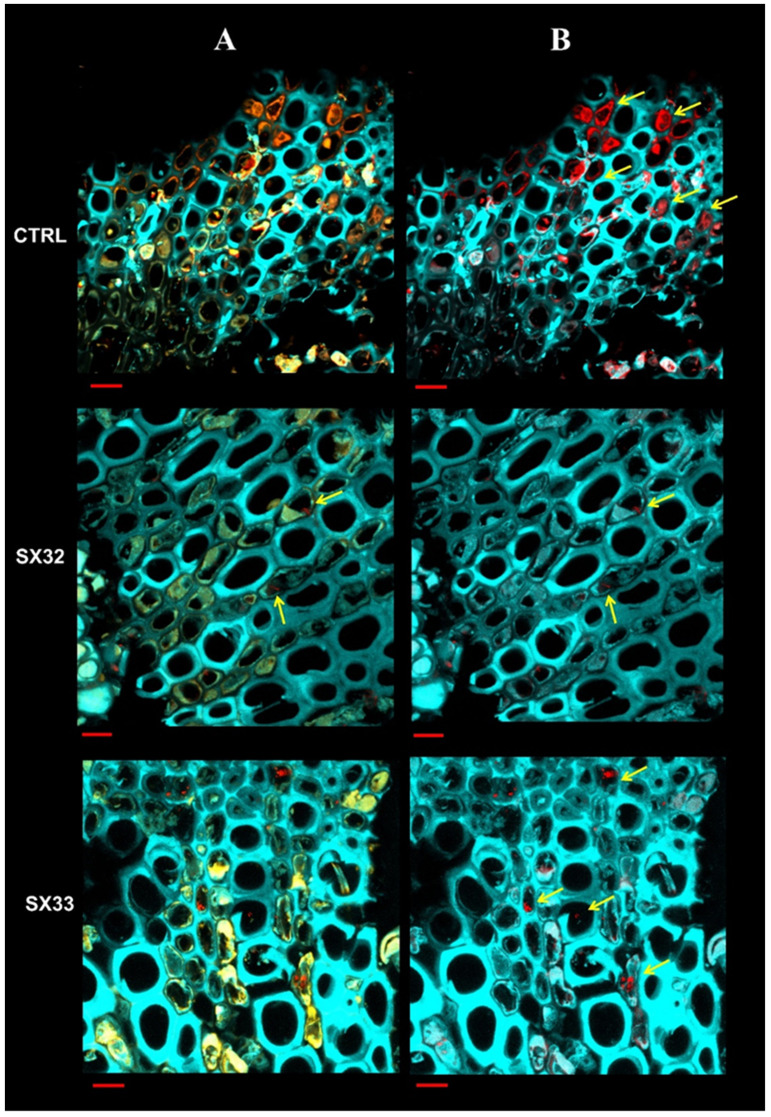
Sections of one-year-old branches representative of control symptomatic (CTRL) and asymptomatic PRPs subjected to FISH staining (fluorescence in situ hybridization) with the KO 210 probe specific for *Xylella fastidiosa (Xf)*. Cyan signal: tissue autofluorescence. Red signal: probe specific for *Xf* labeled with the dye Cy3. Yellow signal: autofluorescence emitted by gels and phenolic compounds. Column (**A**) images obtained by overlapping the three channels: cyan, red and yellow. Column (**B**) images with active the cyan and red channel. *Xf* cells appear either as single cells or as small aggregates often embedded in signal associated with gels and compounds phenolics (yellow arrows). Bar: 10 µm.

**Figure 7 plants-13-00930-f007:**
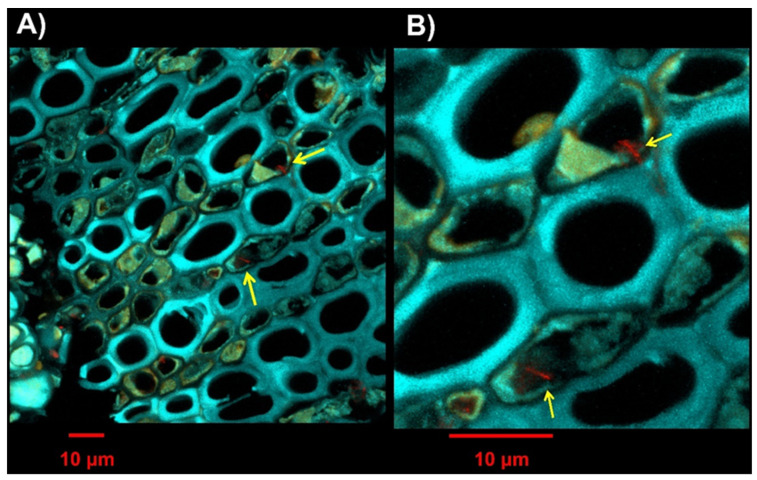
Sections of one-year-old branches of the asymptomatic plant SX_32. (**A**) Cross section with the three acquired canals. (**B**) Detail of image A. Red signal: specific probe for *Xylella fastidiosa* labeled with the fluorochrome Cy3; cyan: plant tissue autofluorescence; yellow: autofluorescence of gels and phenolic compounds. The yellow arrow indicates the presence of *Xf* cells/aggregates.

**Figure 8 plants-13-00930-f008:**
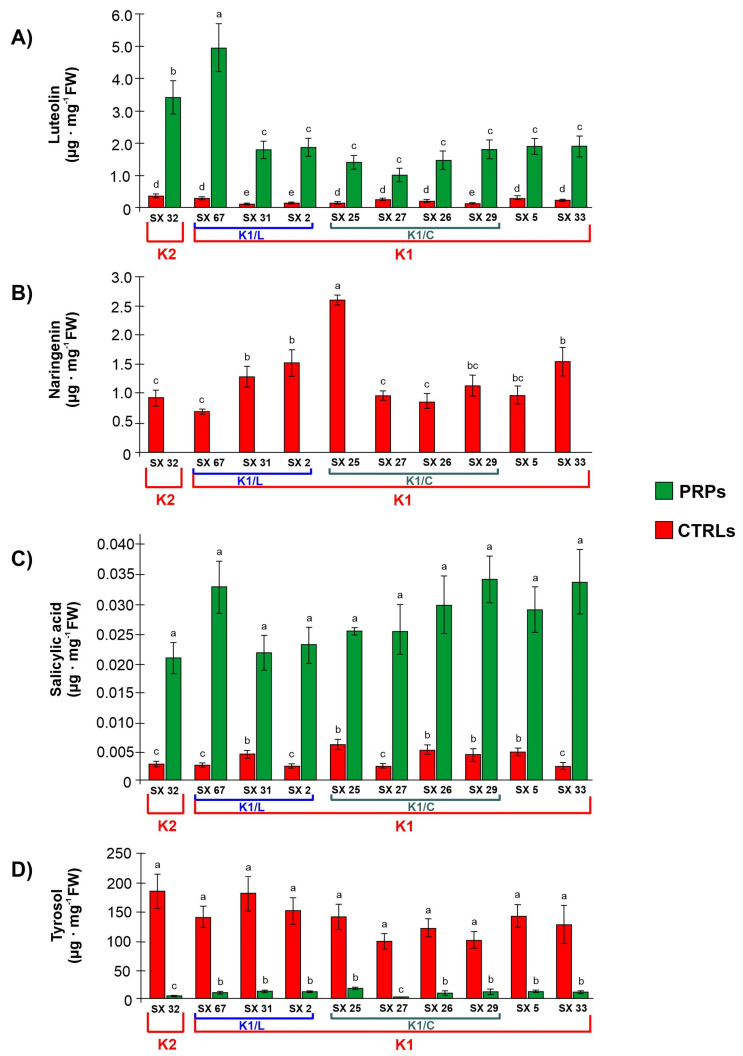
Concentrations of (**A**) luteolin, (**B**) naringenin, (**C**) salicylic acid and (**D**) tyrosol in the stem tissue of *Olea europaea* plants; PRPs: putatively resistant plants; CTRLs: symptomatic plants. According to their genetic profile (assessed by Pavan et al. [12] with a selection of SSR markers), the PRPs are included in the two genetic clusters: K1 (grouping Italian cultivars including Leccino and FS17) and K2 (grouping Tunisian cultivars). The K1 cluster also includes the two sub-clusters K1/L (genotypes closely related with Leccino) and K1/C (genotypes closely related with Ciciulara). These clusters are highlighted in the graph. Different letters indicate statistically significant differences according to ANOVA followed by Tukey-HSD post hoc test.

## Data Availability

Data are contained within the article.

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
