# Peer review of "The Significance of Xylem Structure and Its Chemical Components in Certain Olive Tree Genotypes with Tolerance to Xylella fastidiosa Infection"

_plants, 2024, doi:10.3390/plants13070930_

Round 1
Reviewer 1 Report
Comments and Suggestions for Authors
In the study, the authors investigated the anatomy and histology of xylem vessels, patterns of Xf distribution, and the presence of secondary metabolites in the stems of olive trees between putatively Xf-resistant plants (PRPs) and Xf-infected plants. The findings reveal that the xylem vessel geometry, structural, and chemical defenses may represent a common resistance trait of Xf-resistant plants or genotypes. Overall, this is a well-done study, especially in the presence of secondary metabolites. The findings contribute to the understanding of anatomical and chemical defense in xylem vessels of olive genotypes that are putatively resistant to Xf. However, there are several issues that need to be addressed before publication.
The Introduction needs to be divided into several paragraphs instead of one whole paragraph, as well as Discussion.
Introduction: Since olive is a long-lived tree species and has strong resistance, the major phenolic compounds found in olive should be introduced, such as oleuropein and other secondary metabolites (Tree Physiology, 2024, 44, tpae002).
There are many phenolic compounds that have anti-pathogenic activity, why did you choose luteolin, naringenin, salicylic acid, and tyrosol to study?
The Control (CTRL) olive trees, with a disease severity index ≥1.5, belong to the Xf-susceptible cultivar. The results indicated a significant Xf infection in the CTRL stems, while the PRP stems showed only a mild or subtle infection. Consequently, when comparing CTRL and PRPs, all results displayed significant differences. However, I wonder what the outcomes would be if the CTRL were not infected.
The results indicated a significant accumulation of phenolic compounds in the xylem vessels of PRP. I wonder if this accumulation of phenolic compounds in the putatively resistant olive variety PRP is induced by multiple Xf infections. If PRP is never infected by Xf, will there still be a significant accumulation of phenolic compounds?
Author Response
In the following text, line numbering corresponds to that of the MS file in "Track Change - All markup".
In the study, the authors investigated the anatomy and histology of xylem vessels, patterns of Xf distribution, and the presence of secondary metabolites in the stems of olive trees between putatively Xf-resistant plants (PRPs) and Xf-infected plants. The findings reveal that the xylem vessel geometry, structural, and chemical defenses may represent a common resistance trait of Xf-resistant plants or genotypes. Overall, this is a well-done study, especially in the presence of secondary metabolites. The findings contribute to the understanding of anatomical and chemical defense in xylem vessels of olive genotypes that are putatively resistant to Xf. However, there are several issues that need to be addressed before publication.
- The Introduction needs to be divided into several paragraphs instead of one whole paragraph, as well as Discussion.
- Thank you for your suggestion; Introduction and Discussion sections are now divided into several paragraphs.
- Introduction: Since olive is a long-lived tree species and has strong resistance, the major phenolic compounds found in olive should be introduced, such as oleuropein and other secondary metabolites (Tree Physiology, 2024, 44, tpae002).
- The major phenolic compounds found in olive are introduced by citing the article Tree Physiology, 2024, 44, tpae002 as reported below:
“As long – lived tree species, olive developed the production of specialized metabolites as sophisticated defense mechanisms [4]. In fact, the resistance of plants against vascular pathogens also occurs at biochemical level through the production of secondary metabolites with antimicrobial properties such as phenolic compounds including: secoiridoids (oleuropein), phenolic alcohols (hydroxytyrosol and tyrosol), flavonoids (e.g., apigenin, catechin, epicatechin, kaempferol, luteolin, myricetin, naringin and quercetin), hydroxybenzoic acids (e.g., gallic acid, protocatechuic acid, salicylic acid, syringic acid and vanillic acid), and hydroxycinnamic acids (e.g., coumaric acid, caffeic acid, ferulic acid, p-coumaric acid and sinapic acid) [4, 5, 6].” (lines 68-76).
- There are many phenolic compounds that have anti-pathogenic activity, why did you choose luteolin, naringenin, salicylic acid, and tyrosol to study?
- We only reported the data related to luteolin, naringenin, salicylic acid and tyrosol since they were the phenolic compounds that showed statistically significant differences among the stems of asymptomatic PRPs and symptomatic plants.
- The Control (CTRL) olive trees, with a disease severity index ≥1.5, belong to the Xf-susceptible cultivar. The results indicated a significant Xf infection in the CTRL stems, while the PRP stems showed only a mild or subtle infection. Consequently, when comparing CTRL and PRPs, all results displayed significant differences. However, I wonder what the outcomes would be if the CTRL were not infected.
- CTRL and PRP plants were chosen in olive orchards heavily affected by natural Xf infection. More in detail (as reported in the Material and Method section at lines 324-342), CTRL plants (belonging to the Xf-susceptible cultivar "Cellina di Nardò") were collected in the area in which each PRP has been collected with the aim to represent the Xf infection status of the explored area and so, to give an indication of the inoculum pressure to which the PRPs were subjected. Anymore, not all the obtained results are significant; for example, an anatomical parameter that did not show significant differences in the comparison between CTRL and PRP plants is the vessel density.
The results indicated a significant accumulation of phenolic compounds in the xylem vessels of PRP. I wonder if this accumulation of phenolic compounds in the putatively resistant olive variety PRP is induced by multiple Xf infections. If PRP is never infected by Xf, will there still be a significant accumulation of phenolic compounds?
- As suggested, a possible hypothesis is that these PRPs have a constitutive phenolic composition of the xylem vessels which make them resistant to the Xf infection. In a previous work (Pavan et al., 2021; this work has been cited in the current manuscript with the number [12]) PRPs were also characterized for their genetic profiles (by using a selection of ten SSR markers) and about genetic relations with Italian and Tunisian cultivars. The data will allow additional experiments, in controlled conditions and with artificial bacterial inoculation, to evaluate the constitutive or induced responses to the pathogen infection.
In Conclusions, a sentence has been added as follows:
“Surely, additional experiments in controlled conditions and with artificial bacterial inoculation need to be conducted to evaluate the constitutive or induced responses to the pathogen infection in the olive genotypes here reported”. (lines 471-473)
Reviewer 2 Report
Comments and Suggestions for Authors
See the attached file of comments and suggestions.

This manuscript is composed with a very good English. However, please note the exact usage and meaning of some technical terms.
Author Response
In the following text, line numbering corresponds to that of the MS file in "Track Change - All markup".
- Structural wood-anatomical and chemical defence in xylem vessels of olive genotypes putatively resistant to Xylella fastidiosa
- Spelling; defense.
- Wood and Xylem are not exactly the same. Xylem is a significant structure in the present work.
- Suggested change in the title;
The significance of xylem structure and its chemical components in certain olive tree genotype with tolerance to Xylella fastidiosa infection causing Olive quick decline syndrome
- The title has been changed following the reviewer’s suggestions as follows:
“The significance of xylem structure and its chemical components in certain olive tree genotypes with tolerance to Xylella fastidiosa infection” (lines 2-4).
In addition, “defence” has been changed in “defense” throughout the entire text.
- Lines # 89-91. “simplified the identification of the wood annual rings to proceed with the analysis of the vessel anatomical parameters by using the software WinCell.”
- Xylem is a better term to use than wood.
- This does not present enough justification for including Fig1.
- A further description of the growth, number and sizes of the annual rings supports this figure's inclusion. Fig.1
- Thank you for your suggestions.
“Wood” has been replaced with “xylem” (line 97).
Figure 1 has been removed.
- Figure 1. Representative panorama images (obtained using the PTGui software) of the entire radial section of the branches collected from symptomatic controls (CTRLs) and from the putatively resistant plants (PRPs). • The caption above does not clearly describe the figure content. Which part is the PRPs? define SX !! Cluster !!.
- Needs a statement like this “SXs: representative of the putatively resistant plants (PRPs).”
- It only shows the annul rings, but no xylem details can be see
- As reported in the previous response, Figure 1 has been removed.
- Figure 3. It would be better to make the y-axis values represented by equal distances between the horizontal lines equal (size or distance apart) for both CTRLs and PRPs graphs. As it is now, they are not equal.
- Figure 3 (actually Figure 2) has been modified as suggested.
- Figure 7.
- Most of the xylem vessels in the SXs look larger than the xylem vessels in the PRPs sections?
- The abundance of the bacterial pathogen in that xylem tissue of the CTRLs vs the scarcity in the PRPs may be due to several factors.
- It may be due to period of exposure to the inoculum and its potential and/or suppressive chemical condition.
- This would have been better supported and clarified by artificial inoculation and controlled experimental work using olive material free of this pathogen.
- Thank you for your comment.
SXs is the symbol used to indicate the PRP plants. As suggested, a possible hypothesis is that these PRPs have a constitutive chemical composition of the xylem vessels which make them resistant to the Xf infection. In a previous work (Pavan et al., 2021; this work has been cited in the current manuscript with the number [12]) the analyzed PRPs were also characterized for their genetic profiles (by using a selection of ten SSR markers) and genetic relations with Italian and Tunisian cultivars. These data will allow to conduct additional experiments, in controlled conditions and with artificial bacterial inoculation, to evaluate the constitutive or induced responses to the pathogen infection.
A sentence has been added in the Conclusions section as follows:
“Surely, additional experiments in controlled conditions and with artificial bacterial inoculation need to be conducted to evaluate the constitutive or induced responses to the pathogen infection in the olive genotypes here reported”. (lines 471-473)
- Figure 8.
- This seems part of Fig. 7. What merit does it represent over the preceding figure(s).
- The present description of that xylem ignored other possible structural characteristics and changes such as tylosis and other chemical-structural changes associated with infection?
- As reported in the text, Figure 8 (actually Figure 7) was used to show a detail of the previous images with a higher magnification. The image description reported in the text hypothesized structural changes linked with the tylosis formation as follows:
“An interesting aspect observed in the sections of the PRPs is that the bacterial cells were often included in the autofluorescence signal of gels and phenolic compounds associated with the xylem tissue (Figure 7) and everted into the lumen of the vessel (mechanism that leads to the tylosis formation….)” (lines 180-183).
Discussion
Line # 241. Statement, “SXs cells appeared incorporated in the autofluorescence signal of gels and phenolic compounds associated with the xylem tissue (Figure 8).”
- Needs to be clarified further in contrast with Figure 7.
- What is the scientific logic behind such a statement about a bacterial pathogen travelling in the xylem vessels of the host?
- It would have been further substantiated when the viability of that bacteria was addressed.
-As reported in the previous response, the Figure 8 (actually Figure 7) was used to show a detail of the previous images with a higher magnification. The obtained images have been described by considering the observations achieved in previous published works on other Xf host plant species; of course, as reported in the text, additional experiments will be useful to clarify the host response to the pathogen. A sentence has been added to the discussion section as follows:
“Also for Pierce's disease in grapevine, individual Xf cells or small aggregates have been observed when Xf adheres to the tyloses; in that case, it has been suggested that tyloses, as well as acting as a physical barrier, may supply a desirable surface for Xf colonization, maybe justified from the presence of by-products of cell wall turnover associated with the tylose developing [24].” (lines 253-257).
Lines 242-257. That information belongs to the literature, but it is not directly observed and /or supported by the present findings.
- For the rest of the discussion the chemical aspects were not contrasted to previous published paper(s).
- Ref # 5, Luvisi, A.; Aprile, A.; Sabella, E.; Vergine, M.; Nicolì, F.; Nutricati, E.; Miceli, A.; Negro, C.; De Bellis, L. Xylella fastidiosa subsp. 474 pauca (CoDiRO strain) infection in four olive (Olea europaea L.) cultivars: profile of phenolic compounds in leaves and 475 progression of leaf scorch symptoms. Phytopathol. Mediterr. 2017, 56, 259–273. doi: 10.14601/Phytopathol_Mediterr-20578
- The above reference was only reviewed in the introduction!!
- As reported in the previous response, the discussion at lines 242-257 (actually lines 251-272) has been improved by adding a sentence as follows:
“Also for Pierce's disease in grapevine, individual Xf cells or small aggregates have been observed when Xf adheres to the tyloses; in that case, it has been suggested that tyloses, as well as acting as a physical barrier, may supply a desirable surface for Xf colonization, maybe justified from the presence of by-products of cell wall turnover associated with the tylose developing [24].” (lines 253-257).
Discussion of the chemical aspects was contrasted with the previous published paper (Vergine et al., 2022; reference number [10]) related to the same olive genotypes by adding a sentence as follows:
“These phenolic profiles, obtained by analyzing the stems, showed a different pattern of phenolic composition if compare with the same plants evaluated by processing the leaves in a previous published work [10]; this could suggest a tissue-specific phenolic pattern.” (lines 275-278)
The indicated reference [5] (Luvisi et al., 2017; actually reference [6]) is now cited also in the discussion section at line 266 “….[6, 9, 10]”.